# A Reactive and On-Chip Sensor Circuit for NBTI and PBTI Resilient SRAM Design

**Nandakishor Yadav \***, **Youngbae Kim** , **Mahmoud Alashi** and **Kyuwon Ken Choi**

Department of Electrical and Computer Engineering, 3301 South Dearborn Street, Siegel Hall,
Illinois Institute of Technology, Chicago, IL 60616, USA; ykim102@hawk.iit.edu (Y.K.); malashi@hawk.iit.edu
(M.A.); kchoi12@iit.edu (K.K.C.)
* Correspondence: nyadav6@iit.edu

**Abstract:** Process Variation (PV), Bias Temperature Instability (BTI) and Time-Dependent Dielectric Breakdown (TDDB) are the critical factors that affect the reliability of semiconductor chip design. They cause the system to be unstable and increase the soft error rate. In this paper, a compact on-chip degradation technique using runtime leakage current monitoring has been proposed. The proposed sensor-based adaptive technique compensates for the variation due to PV and aging using the body-bias-voltage-generator circuit. Simulation experiments for three and ten-year stress have been performed. Simulation results proved the superiority of the proposed sensor which provides 33% (up to 0.75 V) more output voltage and 98% sensitivity at 1 V supply voltage compared to the state-of-the-art sensor. The proposed technique mitigates up to 80% PV and BTI effects in SRAM compared to the state-of-the-art techniques.

**Keywords:** CMOS; NBTI; PBTI; Process Variation; SRAM; stability; mitigation; life-time

## 1. Introduction

Constant Voltage Scaling (CVS) is a method that enables device dimensions to decrease deeper into the nanometer scale while maintaining a constant supply voltage. Using this method increases the electric field across the gate oxide (between the channel and the gate oxide). It also causes the occurrence of degradation phenomena such as Negative Bias Temperature Instability (NBTI), Positive Bias Temperature Instability (PBTI), Time-Dependent Dielectric Breakdown (TDDB) and Hot Carrier Injection (HCI) [1,2]. These degradation mechanisms make it too hard to meet the circuit lifetime specification using deep nanometer scaled dimensions. Additionally, the NBTI and PBTI are the critical factors causing degradation bellow 2 nm oxide thickness [2]. Since FinFET devices are not doped, the current density is very high at the channel-gate-oxide interface which causes acceleration of the charge carrier to interface into the gate oxide from the channel and it further increases with the stress [3]. Junctionless-FinFET also has a potential to replace the conventional MOSFET but it also has a problem of BTI and process variation [4]. Hence, it is a primary need to design a robust SRAM memory.

The robustness of the semiconductor memory can be dramatically impacted by the aforementioned degradation mechanisms. SRAM is an on-chip semiconductor memory which can be used as a cache memory along with the multi-core processors. It bridges the speed gap between the logic and the main memory. A microprocessor is a power-hungry unit which transforms the dissipated power into heat which further increases the effect of NBTI that is directly proportional to temperature. BTI becomes crucial due to the thermal runaway. The soft error rate increases in SRAM due to BTI. This error can be reduced by increasing the gate length. Hence, reliability can be improved for high temperature operation [5]. There has been some research done on NBTI resilient SRAM circuit design. In the past, there was no much interest in using on-chip NBTI mitigation techniques since they require

large on-chip area and total power budget. In the modern technology area in not a major issue and production yield is decreasing due to complex chip fabrication process. The on-chip mitigation techniques help to increase the yield and life ot the chip [6]. Singh et al have proposed an oscillator frequency based NBTI sensor which collects the data from the test on-chip PMOS transistor for the specific stress mode [7]. The data is supplied to the measurement mode which decides the recovery mode. The 20 Byte SRAM register was used for stress measurement. The stress circuit consisted of the analog comparator which used fourteen transistors, and the measurement circuit is designed by the NAND Gate based oscillator along with the level converter circuits. The main disadvantage of this sensor circuit is that it produces a nonlinear output with the linear change of the stress time. Furthermore, the output becomes unstable with temperature variations and area overhead.

Sai et al have proposed a multi-path aging sensor [8] which does not require placing a sensor in the longest path that might reduce the performance. This sensor works on the principle of differential multiple error detection which uses an analog differential circuit for stress detection. The main advantage of this sensor lies in its capability of detecting and mitigating the delay fault. Detection of the fault is performed in two or more paths at a time. However, the sensor is nonlinear and has a high area overhead [8]. An impressive NBTI sensor has been proposed for the SRAM register files by Yang et al [9]. This sensor can detect the change in the threshold voltage of the PMOS transistors. Further, the in-situ and in-field technique along with software framework have been used to create the recovery vectors using the measured degraded threshold voltage. However, high computational complexity and the requirement of an off-chip software are the drawbacks of these techniques. Shah et al also proposed an NBTI-based sensor to measure the current change for the SRAM [10]. Despite the compact nature of this sensor, its accuracy and linearity are inferior. In this paper, a new stress measurement sensor for 6T cell based SRAM has been proposed. The proposed sensor monitors the change in the leakage current and converts it into voltage which is then used by a measurement circuit to make a decision and perform the mitigation. Due to its very high input impedance due to diode connected transistors, the sensor is very sensitive towards the change in input leakage current. Two push-pull amplifiers are used with a tuning transistor to give a very high output swing. Sizing of an On-chip transistor can be used to tune the accuracy level of the proposed sensor.

The remainder of the proposed work is as follows. Section 2 discusses the effects of NBTI on the 6T SRAM cell, introduces a circuit-level solution, and explains the remaining research gaps. In Section 3, the proposed stress measurement sensor and simulation results are discussed. The mathematical model of the proposed sensor and analysis of the simulation results are also discussed in this section. The proposed architecture and compensation technique are discussed in Section 4. Section 5 compares the proposed sensor and SRAM architecture to the state-of-the=art sensors circuit. Finally, in Section 6, the conclusion is drawn along with future work.

## 2. Aging Effects on the SRAM

The NBTI and PBTI cause an increase in the threshold voltages of the PMOS and NMOS, respectively, under DC and AC stress following the power-law model. The simplified expression for the DC stress is given by [11] shown in Equation (1)

$$\alpha \Delta |V_{th_{DC}}| = K_{DC} t^n \tag{1}$$

where $n$ is the time constant with value $n = 0.25$ for the molecular hydrogen diffusion, $t$ is the aging time and $K_{DC}$ depends on the technology and material parameters. It also contains the stress and recovery time constants which depend on the material, interfaces trap density, biasing and transistor dimensions. NMOS and PMOS transistors of the SRAM also experience AC stress whose effect is less than that of the DC stress. The AC stress is given by [12]:

$$\Delta |V_{th_{AC}}| \approx \alpha \Delta |V_{th_{DC}}| = \alpha \times K_{DC} \times t^{0.25} \tag{2}$$

where $\alpha$ is the perfection parameter which depends on the operating AC frequency. It has already been testified that the lifetime of the circuit is four times greater when under AC stress than DC stress. The sufficient *ON* time of the transistor in a 6T SRAM cell depends on the clock frequency and the change of the input at the gate terminal. The NMOS and PMOS devices experience DC stress when they are in the ON state and perform recovery when they are in the OFF.

Two principal physical mechanisms were used to analyze the N/PBTI for SRAM: one is contributed by the interface traps and the other uses the deep traps inside the oxide layer. These are modeled as follows [13] for 45 nm and below technology nodes:

$$\Delta Vth, IT \sim exp(-\frac{E_a}{K \times T}) \times \left(\frac{\varepsilon}{t_{ox}}(V_{gs} - V_{th})\right)^{TITCE}$$
$$\times \left(TITFD \times E(V_{gs}, V_{ds})\right) t^{NIT} \tag{3}$$

where *TITCE* is the inversion charge exponent for the interface-trap-inducing threshold voltage degradation, *TITFD* is the oxide electric field dependence for the interface trap inducing threshold voltage degradation, *NIT* is the stress time exponent, *E* is the electric field. A device is known to be at partial-recovery when it is relaxed in which case it reduces the total degradation. This is modeled as follows:

$$\Delta V_{th,AC} = TID0 \times \Delta V_{th} \times exp-TDCD \cdot g \tag{4}$$

where *TDCD* is the channel current exponent for the threshold voltage degradation caused by HCI and the *g* quantity model which depends on the duty cycle of the clock. The HCI effect was modeled to represent the dependence of bias on a wide range of drain-gate voltage [13]. The change in the threshold voltage for the PMOS transistor with respect to various nanometer technologies is shown in Figure 1a. The simulation result shows the quick response in initial stress; then it slows down; hence, in the early age of the chip recovery is required. Figure 1b shows experimental data collected from various published works which also show the quick variation in the initial stage of the chip lifetime as the temperature increases [14].

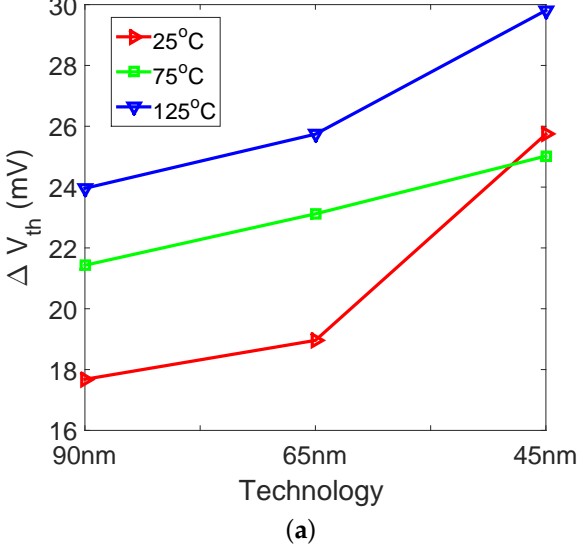

(**a**)

**Figure 1.** *Cont.*

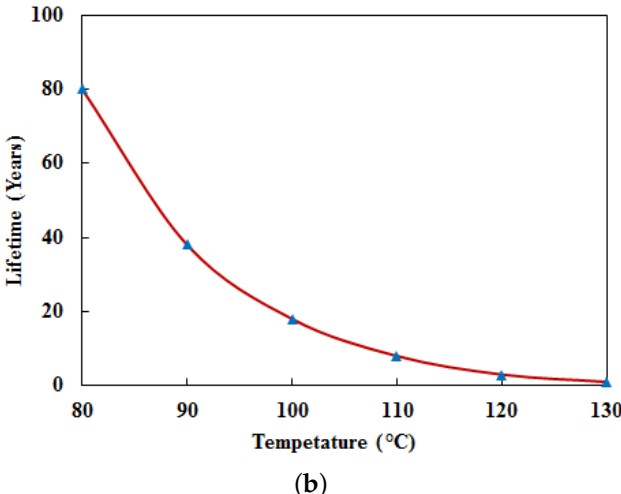

**(b)**

**Figure 1.** (**a**) Threshold voltage variation of PMOS transistor at at stress time of $10^5 s$ [15,16] (**b**) Change in life time with the temperature [14].

Figure 2 shows the basic 6T SRAM cell circuit diagram [17]. PU1 and PU2 are the pull-up transistors, A1 and A2 are the access transistors, PD1 and PD2 are the driver transistors. In the proposed design, a body biasing circuit is used to compensate for the process variation and aging effects. The body terminals of the NMOS and PMOS transistors are separated using the dual-well technology based layouts which are shown in Figure 3. According to Equation (3), the threshold voltage of the pull-up PMOS transistors increases due to NBTI degradation with aging time. The sub-threshold leakage current decreases exponentially as the threshold voltage of PMOS transistors increases with aging time. Accordingly, the total leakage power of SRAM cell decreases with aging time, as shown in Figure 1 for 45 nm technology. However, the increasing value of the threshold voltage of the PMOS transistors in SRAM cell affects the performance and stability. BTI decreases the read SNM, holds SNM, writes margin, word line write margin (WLWM). BTI increases the circuit delay, read and access time failure probability. The read and hold SNM degrades over the aging time under the stressed conditions because of the reduction of the trip-point voltage of the left inverter. The write margin of the SRAM cell gets improved with stress time since the node storing logic "1" (Q) becomes weak and therefore writing becomes easier [18,19].

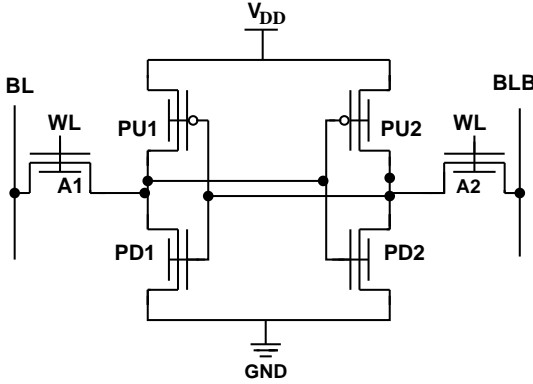

**Figure 2.** 6T SRAM cell.

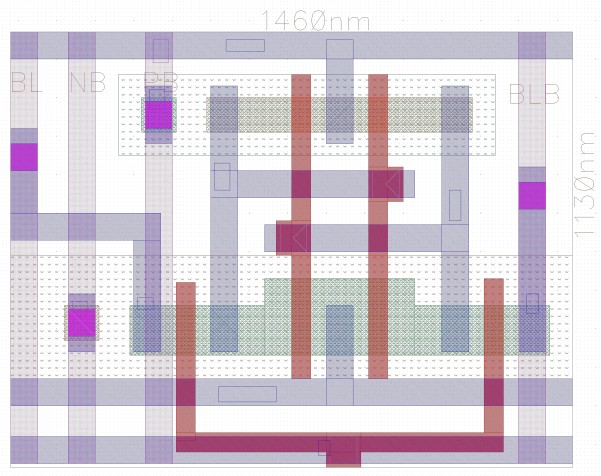

**Figure 3.** 6T SRAM cell layout with separate body terminals.

## 3. Proposed Stress Measurement Sensor

The proposed sensor design is somewhat similar to the design of an analog amplifier. A current to voltage converter based on-chip PV and NBTI sensor circuit has been proposed to compensate for the damages due to PV, BTI (NBTI and PBTI). Figure 4a shows the two-port network of the current to voltage amplifier where $R_{in}$ is the input impedance, $I_i n$ is the input current source, $V_{2Rin}$ is the dependent voltage sour $I_1 Rout$ is the voltage-dependent current source, $R_{out}$ is the output impedance, and $V_{out}$ is the output voltage. The small-signal equivalent circuit of the MOSFET is shown in Figure 4b. This circuit also works like a current to voltage converter circuit as mentioned earlier. However, the input impedance and the gain should be maximized in order to linearize the output swing of the circuit and improve its accuracy.

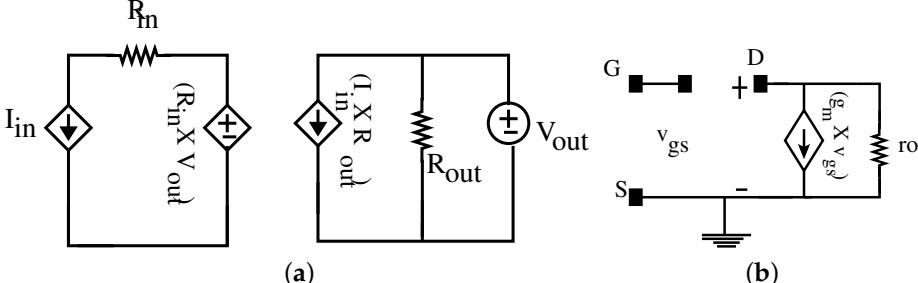

(a)                                                                              (b)

**Figure 4.** (**a**) Two port equivalent network of the proposed circuit. (**b**) Small signal equivalent model of transistor based CCVS.

The modified circuit of the BTI and process variation sensor design is shown in Figure 5. The m1 and m2 transistors convert the current to voltage. Amplification is performed by m3 and m4 transistors. The m1 and m2 transistors are diode-connected and work as a resistor in combination with a voltage device. The m5 transistor works as a tuning device which is used to tune the linearity and output swing of the circuit. We can tune the sensor using M5 transistor width. The layout of the proposed sensor at 45 nm (freePDK) is shown in Figure 6.

Figure 7 shows the simulation based optimization results for M5. It shows the output voltage for different m5 transistor widths and temperatures. It shows an almost linear relationship between the output voltage and the input current when the transistor width is between 140 nm to 170 nm for the input current ranges from 0 to 1 nA. The proposed sensor is tested for 1 KB SRAM memory where column array size is 128 bits (128 SRAM cell is connected in the column). The maximum leakage current is found to have a value of 3 µA for a *FF* process corner. It is clear from Figure 8 that over

a 3 µA range of leakage current, the optimum linear relationship can be achieved by setting $V_{DD}$ = 1.1 V and $W_5$ = 145 nm. This linear range is achieved by increasing the supply voltage. The m6 and m7 transistors constitute the output stage amplifier for the required full output voltage swing. Simulation result of the sensor shown in Figure 9 shows the ability of the proposed sensor circuit to achieve a large change in the output voltage with a small change of the input current. The range of the output voltage swing ranges from 0.1 V to 0.75 V at 1 V supply, which is higher than the output voltage ranges achieved by the state-of-the-art sensor [20]. In order to demonstrate the effect of process variation on the proposed sensor, 5000 Monte Carlo simulations were performed to show the variation in the threshold voltage. The Simulation result is shown in Figure 10 which shows that the mean value µ and the standard deviation (SD) $\sigma$ of the output voltage are smaller than those of the outputs generated by the state-of-the-art designs [20].

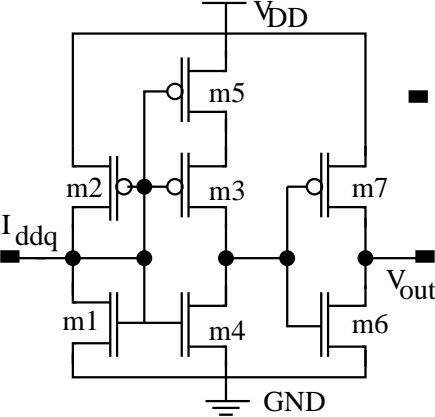

**Figure 5.** Proposed NBTI sensor circuit.

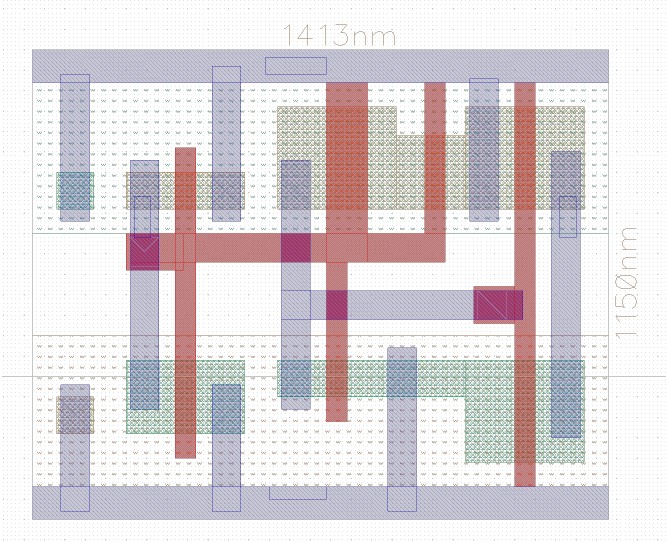

**Figure 6.** layout of the proposed sensor circuit.

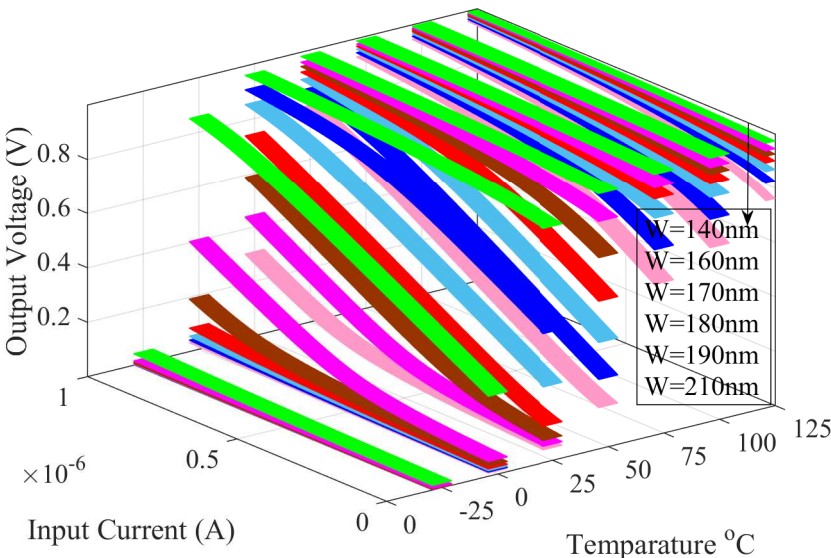

**Figure 7.** Tuning of the sensor circuit using the M5 transistor.

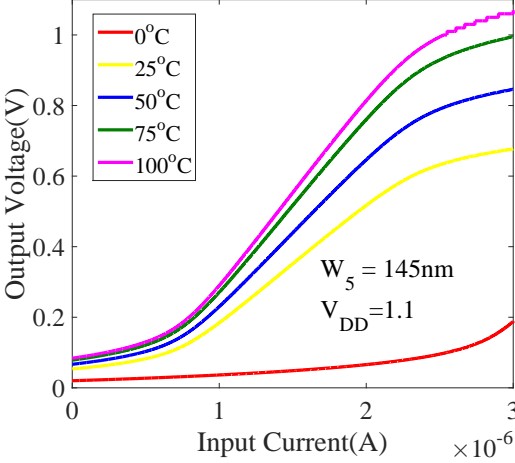

**Figure 8.** Transfer characteristics for the proposed sensor for 3 μ A range.

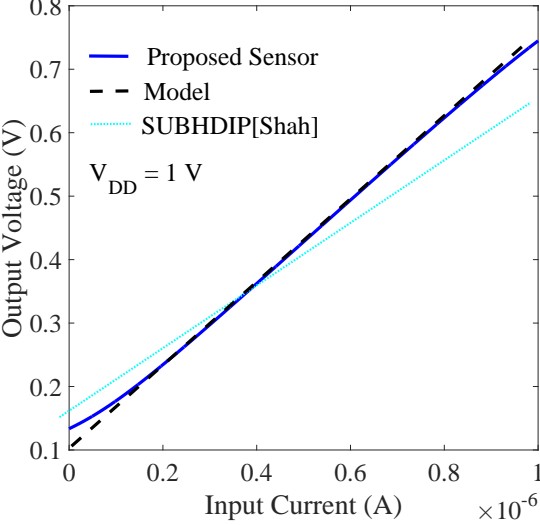

**Figure 9.** Comparative analysis ot transfer characteristics for the proposed sensor.

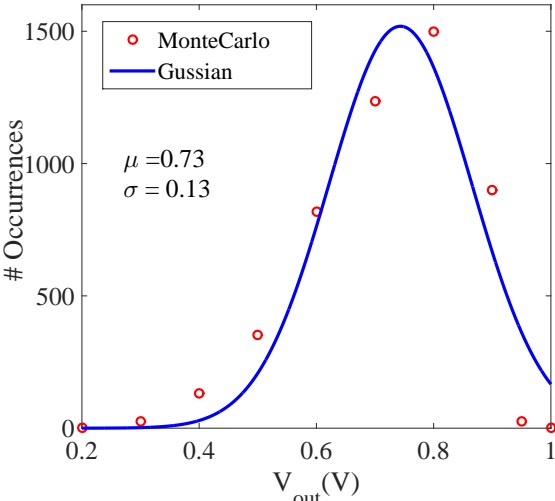

**Figure 10.** Monte Carlo and Gaussian distribution for output voltage of the proposed sensor.

### 3.1. Mathematical Modeling of the Sensor

The physical operation of the sensor can be described by analyzing its electrical equivalent circuit which is shown in Figure 11. The circuit consists of three-stages: The first stage is the current-to-voltage converter circuit which is designed from the diode-connected transistors followed by the variable resistor circuit. $I_{ddq}$ is connected to the gate terminals of the input transistors. Due to the the very high input impedance, $I_x$ is assumed to be zero. Where is $I_x$ shown in the Figure 11.

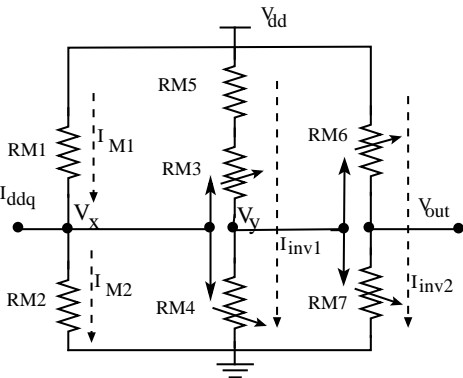

**Figure 11.** Resistor equivalent model of the proposed circuit.

$$IM2 = Iddq + I_{M1} \tag{5}$$

$\therefore$

$$V_x = I_{M2} \times R_{M2} \tag{6}$$

A small change in the $I_{ddq}$ can change $I_{M2}$ which causes a change in the $V_x$ voltage which turns $ON/OFF$ m3 and m4 transistors. Hence, the resistance values of $R_{M3}$, $R_{M4}$, and $R_{M5}$ depend on $V_x$ and $I_{inv1}$ as shown in Equation (7).

$$R_{M3}, R_{M4}, R_{M5} = \int (V_x, I_{inv1}) \tag{7}$$

where $I_{inv1}$ is the current flowing through the first inverter circuit which is followed by the second inverter circuit. Equivalently, $R_{M6}$ and $R_{M7}$ depend on $V_Y$ and $I_{inv2}$ as shown in Equation (8).

$$R_{M6}, R_{M7} = \int (V_y, I_{inv2}) \tag{8}$$

Hence, the output voltage of the sensor circuit is dependent on $I_{inv2}$ and $R_{M7}$. These parameters are also functions of the input current $Iddq$ with an amplification factor.

$$V_{out} = K_M \times I_{inv2} \times R_{M7} \tag{9}$$

where $I_{inv2}$ is the current flowing through the second inverter. This current is a function of $V_x$ which is again a function of $I_{ddq}$ with an amplification factor. $K_M$ is a fitting parameter which depends on the technology used. The above analysis explains the electrical behavior of the proposed sensor circuit. However, the exact value of the input impedance and the response of the output voltage with respect to a small change in the Pico-ampere input current can be only found from the small-signal equivalent model of the sensor circuit.

The simplified small-signal equivalent model of the sensor circuit is shown in Figure 12. It is used to find the transfer function of the proposed sensor. By applying the KCL at the output node, the gain or transfer function of the sensor circuit can be derived as follows:

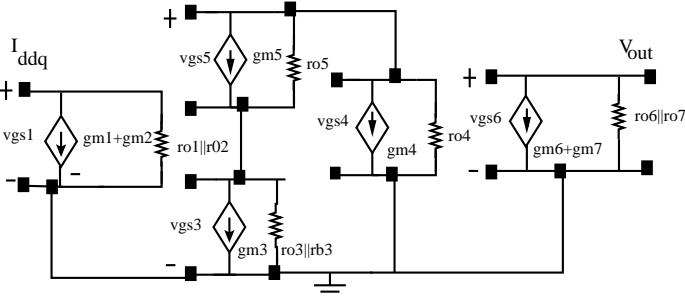

**Figure 12.** Small Signal model of the proposed circuit.

$$\frac{V_{out}}{I_{ddq}} = \frac{(gm3 + gm4 + gm5)(gm6 + gm6)\left[(ro3||ro4) + ro5\right]}{gm1 + gm2 + \frac{1}{ro1||ro2}} \tag{10}$$

$$\frac{V_{out}}{I_{ddq}} = \frac{(gm3 + gm4 + gm5)(gm6 + gm6)\left[(ro3||ro4) + ro5\right]}{gm1 + gm2} \tag{11}$$

If all the transistors have the same dimension and biasing conditions, then the above expression can be simplified as follows:

$$\frac{V_{out}}{I_{ddq}} = 3g_m \times ro \tag{12}$$

where $g_m$ is the trans-conductance and $r_o$ is the output resistance of the transistors. The $V_{out}$ is a function of the output resistance and trans-conductance of the transistors. The sensitivity of the proposed sensor defined by the ratio of the change in output to the change in input as $s = \Delta V_{out}/\Delta I_{in}$, has the value of 0.4 mV/nA which is much higher than the sensitivity of the SUBHDIP sensor [20]. The input resistance in the proposed circuit is very high due to the presence of the gate oxide and a diode-connected transistor. The output resistance should be as small as possible for a good amplifier

design. The output resistance can be calculated at the $V_{out}$ node in Figure 12 by applying a test voltage source and using KCL as follows:

$$\frac{V_{test}}{I_{test}} = \frac{1 - (ro1||ro2)}{(gm6 + gm7) \times \{1 - (ro6||ro7)\}}$$

(13)

A further simplification yields the following:

$$r_{out} = \frac{1}{gm6 + gm7}$$

(14)

where $g_m$ and $r_o$ are known parameters that depend on transistor sizing, threshold, and supply voltage [21]. Figure 9 shows a comparison between the simulation of the proposed sensor circuit and its mathematical model. The simulation results show similar characteristics with the model.

## 4. Proposed Compensation Circuit

The proposed circuit is tested on a 64X1KB butterfly architecture based SRAM where array size is 128B. The compensation circuit takes the input from the sensor circuit and generates a range of back gate voltages for different input voltages. The $V_{BB}$ generator circuit which is used to compensate for the effect of NBTI in the test circuit is shown in Figure 13. The proposed compensation circuit consists of the decision circuit and the back gate/body ($V_{BB}$) voltage generator. The body bias voltage-generator circuit can be designed by connecting the diode-connected MOSFETs in a series which act as series-connected resistances to generate different voltage levels. The sizing of the transistors in the body bias circuit is optimized as discussed in the previous section so that an equal voltage drop is achieved across each diode-connected MOSFET. The voltage drop across each diode-connected MOSFET depends on the output of the amplifier. The VB voltage control circuit is designed using pseudo NMOS logic where *Clock* controls the gate of the transistor $P_k$ which is the weak transistor [22]. The body bias generator circuit can be also designed using an on-chip charge pump [23].

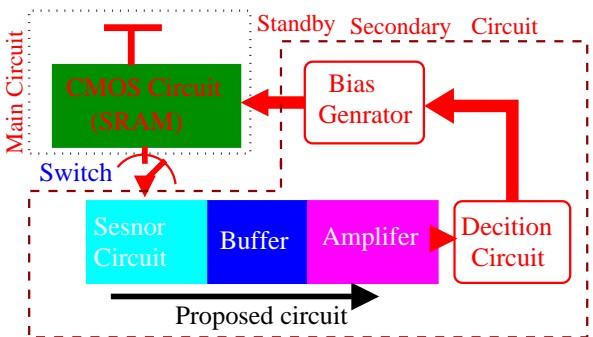

**Figure 13.** Proposed Compensation SRAM architecture.

## 5. Results and Discussion

In this section, the simulation results are discussed under various stress conditions. Simulations were performed using cadence virtuoso and HSPICE tools at 25 °C, 75 °C and 125 °C temperatures. The technology used in the simulations is 45 nm technology PDK. MOS Reliability Analysis (MOSRA) model has already been discussed in section III, and is used for the HCI, NBTI, and PBTI induced stress analysis [24].

The mico-chip layout of the butterfly based SRAM architecture is shown in Figure 14 with precharge circuits on the top. SRAM array, decoder, and sensor circuits are connected with all the arrays where SRAM array depth is 128B. The multiplixer and sense amplifier are connected bellow the

sensor circuits. The total area of the chip without *IO* pad is 387 × 183 μm and the area occupied by the sensor circuit is 374.1 × 1.15 μm which is 0.6% of the total area.

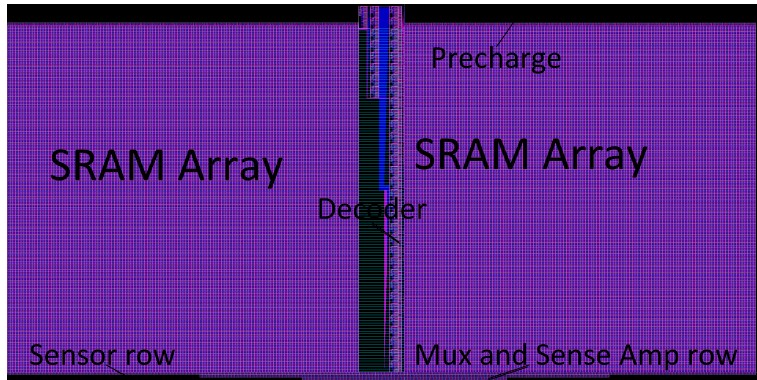

**Figure 14.** Butterfly architecture based without IO-pad SRAM microchip.

Usually in 6T SRAM, more reading are performed than writing operations. So, the change in the threshold voltage of the 6T SRAM transistors has been calculated in the reading mode as shown in Figure 15. It can be shown from the results that maximum stress is achieved on the driver transistor. Since the Access transistors are directly connected with the wordline and bitlines, they are also affected by the PBTI stress more than the drive transistor. The standby leakage current changes with a change in stress for the SRAM cell. Such a change of the leakage current is measured by a proposed NBTI sensor. The same sensor can be also used to mitigate the effects due to process variation. The read time (RT), write time (WT), and the leakage current for various process corners for 128B SRAM array are shown in Table 1. The minimum and maximum values of the leakage current are 27 nA and 3.76 uA, respectively. Hence we tune the sensor for 3 nA current range as shown in Figure 8. In order to find the body biasing voltage which is required to shift back the threshold voltage to its normal value as in TT corner, write and read time simulations are performed for SS and FF to TT process corner using different body biasing voltages. Some of the simulation results for SS process corner are shown in Table 2 where it can be seen that the normal threshold voltage at the TT corner can be restored by applying the highlighted body voltages. Similarly, the desired voltages were found from the FF process corner simulations to be ($BB_{PMOS}$ = 1.8 V and $BB_{NMOS}$ = −0.8 V). The range of the body biasing voltage can be generated using the proposed sensor and charge pump circuits since the range of the proposed sensor output voltage is 0.1 V to 1 V.

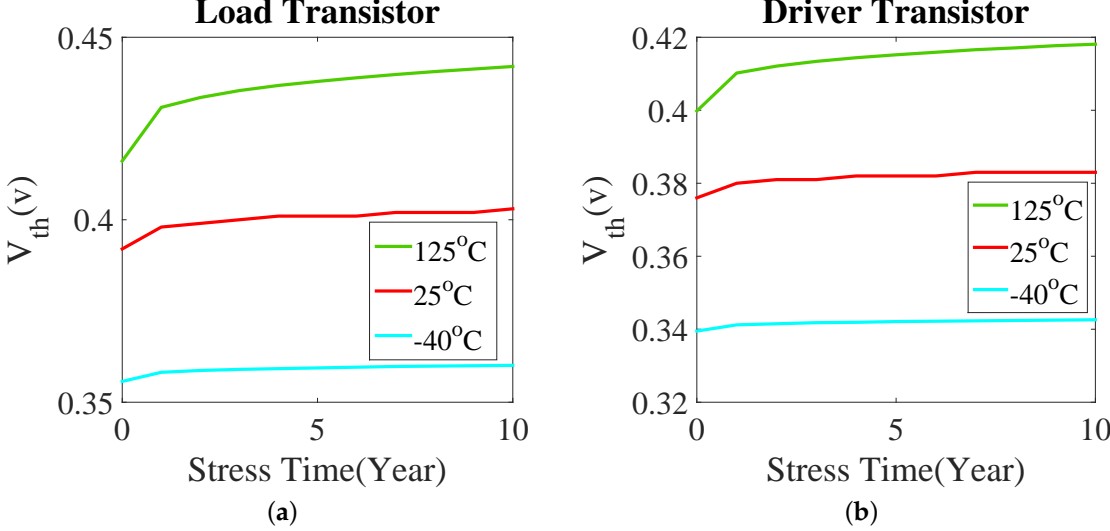

**Figure 15.** *Cont.*

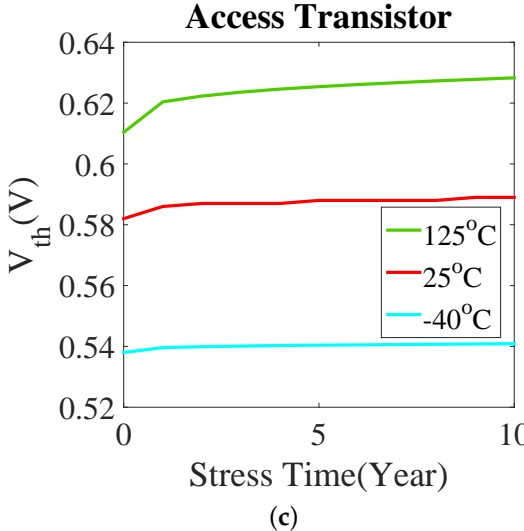

(c)

**Figure 15.** Degradation in $V_{th}$ of SRAM cell transistors in 10 year NBTI & PBTI stress.

**Table 1.** Process Corner and leakage currents.

| Process Corners | Write Time(s) | Write Time(s) | Leakage Current($A$) |
|---|---|---|---|
| TT | $5.81 \times 10^{-11}$ | $9.51 \times 10^{-11}$ | 475n |
| FF | $4.59 \times 10^{-11}$ | $7.61 \times 10^{-11}$ | 3.76u |
| FS | $6.01 \times 10^{-11}$ | $7.83 \times 10^{-11}$ | 2.68u |
| SF | $9.33 \times 10^{-11}$ | $1.72 \times 10^{-10}$ | 1.13u |
| SS | $1.22 \times 10^{-10}$ | $2.08 \times 10^{-10}$ | 27.9n |

**Table 2.** Body biasing voltage.

| Read Time(s) | Write Time(S) | $BB_{PMOS}$ | $BB_{NMOS}$ |
|---|---|---|---|
| $2.08 \times 10^{-10}$ | $1.22 \times 10^{-10}$ | 1.0 | 0.0 |
| $2.09 \times 10^{-10}$ | $1.20 \times 10^{-10}$ | 0.95 | 0.05 |
| $2.04 \times 10^{-10}$ | $1.19 \times 10^{-10}$ | 0.9 | 0.1 |
| $2.06 \times 10^{-10}$ | $1.18 \times 10^{-10}$ | 0.85 | 0.15 |
| $1.99 \times 10^{-10}$ | $1.18 \times 10^{-10}$ | 0.8 | 0.2 |
| $1.96 \times 10^{-10}$ | $1.16 \times 10^{-10}$ | 0.75 | 0.25 |
| $1.96 \times 10^{-10}$ | $1.14 \times 10^{-10}$ | 0.7 | 0.3 |
| $1.95 \times 10^{-10}$ | $1.13 \times 10^{-10}$ | 0.65 | 0.35 |
| $1.94 \times 10^{-10}$ | $1.14 \times 10^{-10}$ | 0.6 | 0.4 |
| $1.94 \times 10^{-10}$ | $1.13 \times 10^{-10}$ | 0.55 | 0.45 |
| $1.95 \times 10^{-10}$ | $1.13 \times 10^{-10}$ | 0.5 | 0.5 |
| $1.91 \times 10^{-10}$ | $1.12 \times 10^{-10}$ | 0.45 | 0.55 |
| $1.93 \times 10^{-10}$ | $1.12 \times 10^{-10}$ | 0.4 | 0.6 |
| $1.88 \times 10^{-10}$ | $1.14 \times 10^{-10}$ | 0.35 | 0.65 |
| $1.90 \times 10^{-10}$ | $1.16 \times 10^{-10}$ | 0.3 | 0.7 |
| $1.91 \times 10^{-10}$ | $1.20 \times 10^{-10}$ | 0.25 | 0.75 |
| $1.95 \times 10^{-10}$ | $1.28 \times 10^{-10}$ | 0.2 | 0.8 |

The standby leakage current of SRAM cell at 45 nm technology is known to be around a few nano-amperes [25–27]. RNM and SNM are the read and static noise margins, respectively, which are the parameters used to examine the stability of the SRAM cell. The performance of SRAM is examined using the word line write margin (WLWM) and read current parameters. Figures 16 and 17 show the degradation of the RNM and SNM due to NBTI at 25 °C and 125 °C temperatures. The effect of NBTI further increases at higher temperatures where it further degrades the stability of the SRAM.

Figure 18 shows the degradation of the WLWM under NBTI stress at 25 °C and 125 °C temperatures, respectively. It is clear that the WLWM is more degraded by the NBTI effect than the stability parameters. The amount of degradation of WLWM also increases at higher temperatures. NBTI is a more critical issue in PMOS than PBTI. However, a combination of NBTI and PBTI which exists in most of the real cases has a much bigger effect on the stability and performance of the SRAM cell than the effect due to NBTI only. In addition, the combined effect of NBTI and PBTI causes the change of the threshold voltage of the transistors. The high impact of the combined effect of NBTI and PBTI on the design is also demonstrated by the variation in WLWM as shown in Figure 19. The effects of NBTI, PBTI, and their combination on reading, writing, and the standby leakage current at 25 °C temperature are shown in Table 3.

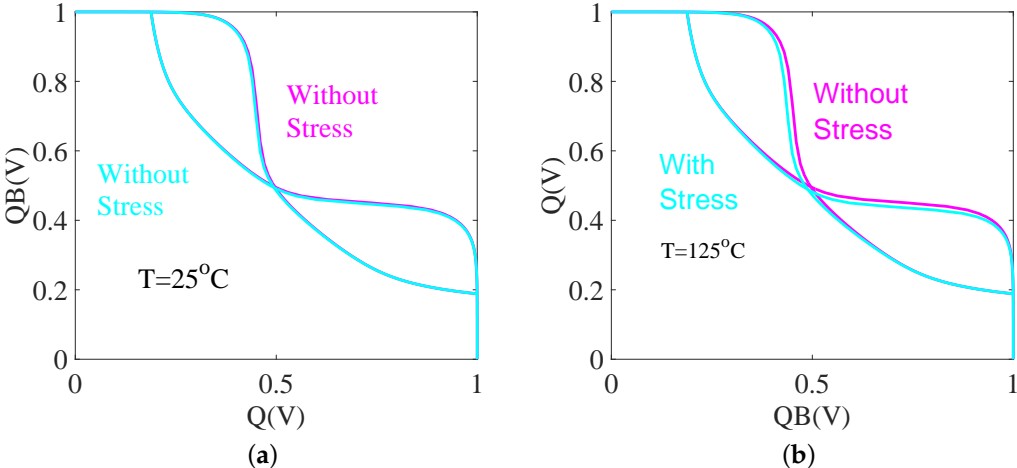

**Figure 16.** Degradation in RNM under 10 year NBTI stress at (**a**) *T* = 25 °C. (**b**) *T* = 125 °C.

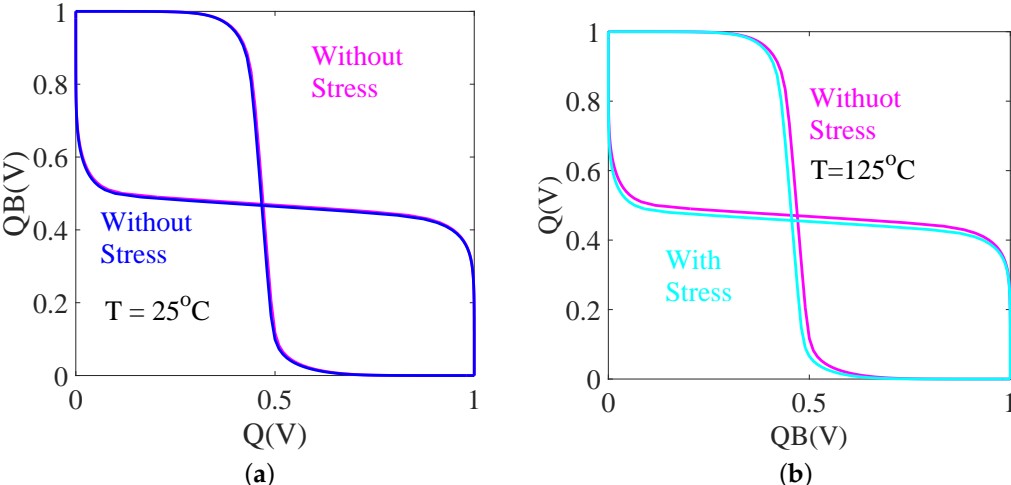

**Figure 17.** Degradation in SNM under 10 year NBTI stress at (**a**) *T* = 25 °C. (**b**) *T* = 125 °C.

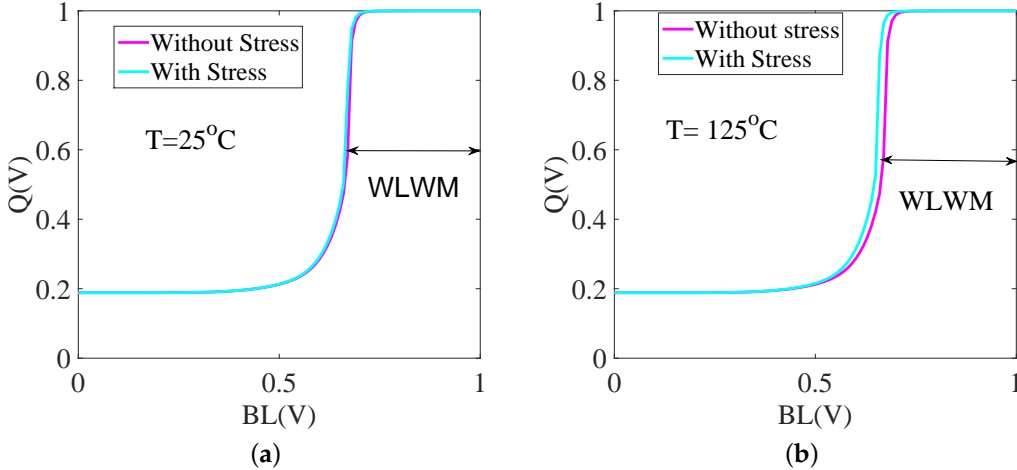

**Figure 18.** Degradation in WLWM under 10 year NBTI stress at (**a**) *T* = 25 °C. (**b**) *T* = 125 °C.

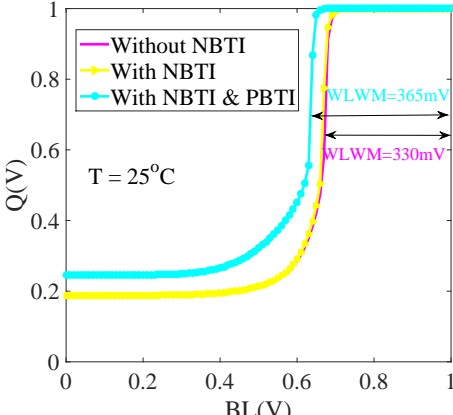

**Figure 19.** Degradation in WLWM under 10 year NBTI and PBTI stress.

**Table 3.** Degradation in SRAM stability and performance with 10 Year stress.

| Parameters (for 10Y Stress) | SNM (mV) | WLWM(mV) |
|---|---|---|
| Without Stress | 361 | 330 |
| with NBTI | 310 | 345 |
| With NBTI and PBTI | 187 | 365 |

Figure 20 shows a comparison between the proposed sensor-based mitigation technique and that of the state-of-the-art under the combined NBT and PBTI stress. The effectiveness of the proposed O-ABB circuit based mitigation technique has been examined by comparing simulations performed for the SRAM cell with and without the use of the O-ABB circuit. The body-bias voltage generator circuit provides different output voltages with different stress value. The precision of this circuit depends on the input voltage provided by the sensor. Therefore, it is linear and sensitive towards a minimal change in the input current.

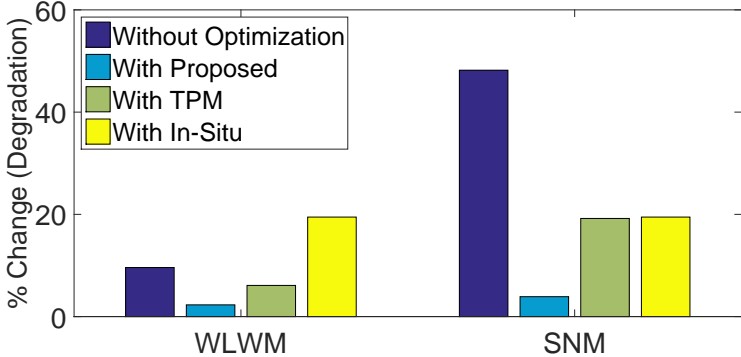

**Figure 20.** Comparison of stability and write performance failure for proposed and existing techniques In-Situ [9] and (TPM) [28].

The proposed design has many advantages but every design also has limitations in terms of speed, area and stability. The proposed sensor can perfectly work for the temperature range 0 °C to 100 °C. However, sensor can be made work in a wider range of temperatures by increasing the six=ze of the transistors. The proposed design also requires larger design area and higher power budget. One of the key limitations of this circuit is that its required minimum working supply voltage is 1.1 V. The additional clock cycle needed for the proposed circuit causes delay and limits the performance. Therefore, Improving the circuit design to allow its operation at lower voltage levels without additional clock cycles will be our future research topic.

## 6. Conclusions

The process variation, BTI (NBTI and PBTI) are the most critical issues affecting the reliability of the circuit design. They become more critical for systems designed for defense and aerospace applications because these reliability degradation sources decrease the system's stability and life. Mostly, SRAM is used as cache memory inside the microprocessor to bridge the gap between the logic and the main memory speed. The shift in the threshold voltages of SRAM cells due to BTI can cause degradation of the performance and stability. The proposed BTI mitigation technology not only detects small changes in the device threshold voltage and mitigates the consequent degradation, but also detects the change in the leakage current and converts it to voltage. Simulation results show that the proposed sensor provides 33% higher output voltage and 98% sensitivity at 1V supply voltage than other conventional state-of-the-art sensors. In addition, the proposed technology can mitigate up to 80% of the effects due to NBTI and PBTI in SRAM compared to conventional technologies. The output voltage has been calculated using the small-signal equivalent model of the sensor circuit and was compared with the simulation result. The simulation of the model also proved the efficiency of the proposed sensor. The area overhead of the proposed techniques can also be decreased by increasing the SRAM size. The proposed sensor circuit can be used with any analog and digital CMOS circuit.

**Author Contributions:** Conceptualization, N.Y.; Investigation, Writing-original draft, Writing-review and editing, Y.K.; Writing draft and editing, M.A.; Editing, K.K.C.; Funding acquisition and supervisor. All authors have read and agreed to the published version of the manuscript.

**Funding:** "This research was funded by the Industrial Core Technology Development Program of MOTIE/KEIT, KOREA grant number 10083639."

**Acknowledgments:** We thank our colleagues from KETI and KEIT who provided insight and expertise that greatly assisted the research and greatly improved the manuscript. This work is also supported by the Industrial Core Technology Development Program of MOTIE/KEIT, KOREA [# 10083639.]

**Conflicts of Interest:** The authors declare no conflict of interest.

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
