# Peer review of "A Reactive and On-Chip Sensor Circuit for NBTI and PBTI Resilient SRAM Design"

_electronics, doi:10.3390/electronics9020326_

Round 1

Reviewer 1 Report

1. Regarding the circuit simulation in Figure 5, Figure 7 and 8 only shows the output voltage versus input current at one temperature. Since in reality, the temperature is not fixed, please show how the curves in Figure 7 and 8 change with both temperature and input current in a 3-D plot figure.

2.  In Section 4, the authors state that "Simulations were performed using cadence virtuoso and HSPICE tools at 25C, 75C and 125C temperatures." I think you should consider low temperatures such as 0C and -40C as well. Please provide simulation results of 0C and -40C.

3. Please provide more design details of "Bias Generator" circuits in your Figure 13.

4. In Figure 20, the authors only compare the results with [24], which is not sufficient. The references [4]-[6] should be added to Figure 20 for discussion and comparison.

5. Moreover, the authors should mention that the overall system power supply may be limited by the proposed sensor circuit in Figure 5. This is a potential disadvantage of this design. For example, the supply voltage of the sensor circuit is 1V for 45nm technology. Other low power SRAM designs may run at even lower supply voltages, such as

[1] A. Banerjee, B. Calhoun, "A double pumped single-line-cache SRAM architecture for ultra-low energy IoT and machine learning applications," International Conference on VLSI Design, pp. 299-304, 2019.

[2] X. Wang, C. Lu, Z. Mao, "Charge recycling 8T SRAM design for low voltage robust operation," International Journal of Electronics and Communications, vol. 70, pp. 25-32, 2016.

[3] V. Sharma, M. Gopal, P. Singh, S. Vishvakarma, "A 220mv robust read-decoupled partial feedback cutting based low-leakage 9T SRAM for the Internet of Things (IoT) applications", International Journal of Electronics and Communications, vol. 87, pp. 144-157, 2018.

[3]

Reviewer 2 Report

My main concern is about the provided literature, which is outdated.

Therefore, I suggest to the authors to perform an in-depth literature survey in order to include all the published material and include that in the comparison Table.

Another comment is that the limitations of the proposed scheme must be discussed.

Round 2

Reviewer 1 Report

The quality of this work has been improved

Reviewer 2 Report

My concerns have been adequately addressed by the authors.